# Taxonomic characterization and cytotoxic potential of Vietnamese *Ganoderma ellipsoideum* against human breast cancer MCF-7 cells

Trang Thi Thu Nguyen[1,2], Phu Tran Vinh Pham [3], Viet An Thi Nguyen[4], Trinh Thi Tuyet Nguyen[1,2], Linh Thuoc Tran[1,2], Phu Chi Hieu Truong[5], Manh Hung Tran [5*]

**1** Faculty of Biology and Biotechnology, University of Science, Ho Chi Minh City, Vietnam, **2** Vietnam National University, Linh Trung, Thu Duc City, Ho Chi Minh City, Vietnam, **3** VNUK Institute for Research and Executive Education, The University of Danang, Hai Chau, Danang City, Vietnam, **4** University of Science and Technology of Hanoi (USTH), Vietnam Academy of Science and Technology, Nghia Do, Hanoi, Vietnam, **5** School of Medicine and Pharmacy, The University of Danang, Ngu Hanh Son, Danang City, Vietnam

\* tmhung@smp.udn.vn

## Abstract

This study reports for the first time the occurrence of *Ganoderma ellipsoideum*, a wood-decaying fungus, in Vietnam. Species identification was achieved through morphological characterization and sequencing of the internal transcribed spacer region of nuclear ribosomal DNA. Phylogenetic analysis confirmed that the Vietnamese specimens clustered within the *Ganoderma ellipsoideum* taxon, supported by high bootstrap and posterior probability values (90%/1.00). Morphological features further indicated its placement within the *Ganoderma applanatum–australe* complex. *In vitro* cytotoxicity assays revealed that the ethanol extract and its sub-fractions (n-hexane, ethyl acetate, aqueous) exerted inhibitory effects on human breast cancer MCF-7 cells, with the ethyl acetate fraction showing the strongest activity. *In silico* molecular docking demonstrated strong binding affinities between major triterpenoid compounds and key breast cancer-related proteins, including HPA, MELK, CK2α, and NUDT5. These findings not only establish *Ganoderma ellipsoideum* as a newly recorded species in Vietnam, but also suggest its promising potential as a source of anticancer agents.

## Introduction

The application of medicinal herbs and mushrooms for the treatment and prevention of diseases has been a longstanding component of traditional medicine across various countries and regions worldwide [1–3]. Among the most dangerous and chronic diseases in humans, cancer is increasing dramatically each year and becoming the leading cause of death [4]. Nowadays, most cancers are treated by taking drugs, chemotherapy, or radiation therapy after surgery [5,6]. Unfortunately, these therapies

**Data availability statement:** All data are in the manuscript.

**Funding:** The authors would like to thank the Management Board of Ea So Nature Reserve, Eakar District, Dak Lak Province for providing mushroom samples for this study. Trang Thi Thu Nguyen was funded by the Vingroup Joint Stock Company and supported by the PhD Scholarship Program of the Vingroup Innovation Foundation (VINIF), Vingroup Big Data Institute (VINBIGDATA) (grant number: code VINIF.2020.TS.68). The funders had no role in study design, data collection and analysis, decision to publish, or preparation of the manuscript.

**Competing interests:** The authors have declared that no competing interests exist.

are often toxic to patients because of their side effects [6,7]. Therefore, the discovery and development of new treatment solutions based on knowledge of traditional medicine in using medicinal plants to prevent, cure, and recover cancer treatment patients is growing strongly worldwide. Among cancer diseases in humans, breast cancer, which accounted for 14% of all cancer fatalities and 23% of all new cancer cases in 2008 [8], is the most common cancer mortality and the leading cause of cancer death in women globally with 650 000 deaths, 1 in every 6 cancer deaths in women in 2020 [9].

Among natural herbs, *Ganodermataceae* species have essential pharmacological properties such as anticancer [10–12], especially anti-breast cancer [13]. Many reports indicated the extracts of *Ganodermateace* species such as *Aumauroderma rude* [14], *Ganoderma applanatum* [15], *G. cochlear* [16], *G. lucidum* [17–19], *G. leucocontextum* [20], *G. multiplicatum* [21], *G. martinicense* [21], *G. resinacum* [22,23], *G. sinense*, *G. tsugae* [24] exhibited anti-breast cancer properties. Among the *Ganodermateace* species, *G. ellipsoideum* was the first discovered on Hainan Island, China [25]. In 2021, Luangharn and colleagues described phylogenetic and morphological analysis of 22 *Ganoderma* species collected from the Greater Mekong Subregion, in which there was a new record of *G. ellipsoideum* from Thailand [26]. Sappan et al. reported lanostane-type triterpenoids (ganoellipsic acids) from artificial cultivation of fruiting bodies by using the phenylglycine methyl ester method [27]. However, information regarding *Ganoderma elipsoideum*, particularly its morphology and biological functions, including anticancer properties in specimens collected from Vietnam remains limited and outdated.

Breast cancer is recognized as the leading cause of death in women worldwide and female breast cancer became the most frequently diagnosed cancer with about 2.3 million new cases in 2022 globally [28]. Accordingly, new drugs to combat breast cancer cells and regulate this disease are urgently required. Breast carcinoma is closely related to some gene products such as Helix Pomatia Agglutinin (HPA), Maternal Embryonic Leucine Zipper Kinase (MELK), Human Protein Kinase 2 alpha (CK2a), and NUDIX hydrolase 5 (NUDT5). HPA is a useful marker for primary breast cancer diagnostics. In breast cancer cells, HPA-binding properties cells have been correlated with the migration of cells to secondary sites. The migration of cancer cells to secondary sites by favoring the binding of cancer cells to endothelial and the transmigration to the organ of metastasis may be facilitated by the alterations in cellular glycosylation [29]. In triple-negative breast cancer, MELK is abundant; therefore, it is a promising candidate for molecular diagnostics and therapy. MELK is a serine/threonine kinase protein belonging to the AMP-activated protein kinase-associated kinase family, it is a crucial molecule in tumor cell cycle regulation and growth signaling pathways [30]. CK2a is one of the protein kinases involved in cell growth and proliferation of cancers. Therefore, small molecule inhibitors of CK2a have been explored such as peptide-based therapy and the use of CK2a antisense and RNAi to get molecular downregulation of CK2a [31]. NUDT5 is a nucleotide-metabolizing enzyme that is involved in the metabolism of ADP-ribose and 8-oxo-guanine. Therefore, NUDT5 has been identified as a key factor in ATP production in the nucleus of

breast cancer cells. Moreover, because of the important function in the action of estrogen, NUDT5 could be a novel drug target and biomarker for the treatment of ER-positive breast cancer [32]. Due to the biological functions of those mentioned proteins in breast cancer cells, they are considered as drug targets for breast cancer treatment.

Therefore, this study was designed to identify and classify the *Ganoderma ellipsoideum* VNES22015-A1 collected in Vietnam using both morphological characteristics and molecular phylogenetic analysis based on ITS rDNA sequencing, and evaluate the potential anticancer activity of its ethanol extract and solvent-partitioned fractions against human breast cancer MCF-7 cells using *in vitro* cytotoxic assays and *in silico* molecular docking against key breast cancer-related targets (HPA, MELK, CK2α, and NUDT5).

## Materials and methods

### Sample collection and chemical reagents

Mushroom specimens were collected in Ea So Nature Reserve, Dak Lak Province at 108° 36'13.8" E and 12°54'11.4" N, 279 m elevation, temperature 27.6°C and humidity 91% on 27th November 2022. Human breast cancer adenocarcinoma (MCF-7) cell line was obtained from the Department of Genetics, Faculty of Biology and Biotechnology, University of Science, Vietnam National University Ho Chi Minh City. Chloroform, dimethyl sulfoxide (DMSO), ethyl acetate, ethanol, *n*-hexane, isoamyl alcohol, and phenol were purchased from Merck KgaA (Darmstadt, Germany). Potato dextrose agar (PDA) was from HiMedia (Mumbai, India). Eagle's minimal essential medium (EMEM) and fetal bovine serum (FBS) were purchased from Gibco (Grand Island, NY, USA). Amphotericin B, L-glutamine, camptothecin, ciprofloxacin, penicillin G, streptomycin, sulforhodamine B (SRB), trichloroacetic acid were obtained from Sigma (St. Louis, MO, USA). Other chemicals used for DNA extraction and PCR amplification were obtained from Thermo Fisher Scientific (Waltham, MA, USA).

### Morphology analysis

Macroscopic features of collected mushroom specimens were observed by using a Canon EOS 7D (Ōta, Tokyo, Japan) camera on fresh fruiting bodies. Colour codes from Kornerup & Wanscher were used for identifying the colour standards [33]. The basidiomes were cut into small pieces, inserted into a 5% KOH solution, stained with a 1% cotton blue and Melzer reagent solution, and observed by a stereomicroscope (Meiji EMZ 127212, Saitama, Japan). Fruiting bodies were observed at 1000× magnification using an optical microscope (S/N EU1611034 microscope and CMEX-10 PRO USB 3.0 microscope camera, Euromex, Arnhem, Netherlands). The size of at least 50 basidiospores was measured. Mushroom samples were dried at 50°C to a consistent weight and stored in zip plastic bags with silica gel. The voucher specimens were deposited at the Laboratory of Microbiology, Faculty of Biology and Biotechnology, University of Science, Vietnam National University Ho Chi Minh City, Vietnam. The mycelia were isolated by transferring fresh tissue of basidiocarps to PDA medium with ciprofloxacin (50 µg/mL) in Petri dishes under sterile conditions and cultivating them at 25±2°C for 10 days. Mycelia were stored at 4°C for future study.

### DNA extraction, PCR amplification, and DNA sequencing

Total DNA of mycelia samples was extracted and purified using the phenol:chloroform:isoamyl alcohol technique with TE buffer (Tris HCl pH 8.0, 1.0 mM EDTA) in the presence of RNAse A (60 µg) [34]. The DNA was evaluated for quality and concentration by Nanodrop 2000 Spectrophotometer (Thermo Fisher Scientific, USA), then was held in a Sanyo biomedical freezer at −30°C for long-term storage. A specific pair of primers ITS1F (5'-CTTGGTCATTTAGAGGAAGTAA-3') and ITS4 (5'-TCCTCCGCTTATTGATATGC-3') were used to amplify the internal transcribed spacer (ITS) regions of nuclear rDNA [35]. A total volume 12.5 µL of Polymerase cycle reaction (PCR) master mix including 2 µL sample DNA, 6.5 µL ReadyMix™ Taq PCR Reaction Mix, 0.25 µL (100 µM) each primer, and 4.75 µL double-distilled water. The PCR thermal cycle included a 2-minute initial denaturation step at 98°C followed by 35 cycles of denaturation at 98°C for 15 seconds

(s), primer binding at 55°C for 30 s, elongation at 72°C for 1 minute, and the final extension step at 72°C for 1 minute in a Mastercycler Pro (Eppendorf, Germany). PCR products (5 μL) were checked on a 1% agarose gel electrophoresis in Tris-acetate EDTA buffer stained with ethidium bromide, then observed and photographed in the UV lightbox. The amplified PCR products were purified by PureIT ExoZAP PCR CleanUp (Ampliqon, Odense M, Denmark). The purified PCR products were then sequenced by 1st BASE Laboratories (Apical Scientific, Singapore) using Sanger method. DNA sequence after processing was submitted on Genbank (https://www.ncbi.nlm.nih.gov/) with accession number OQ780769.

## Phylogenetic analysis

The ITS sequences of mushroom species were assembled with the software BioEdit 7.2 and modified manually. The newly obtained sequence has been submitted to GenBank using Basic Local Alignment Search Tool (BLAST). Results of related genera on studies or searched and downloaded from the GenBank database (Table 1) were acquired using 5.8S

**Table 1. Sequences of *Ganoderma applanatum-australe* complex species used in this study.**

| Species | Genbank No. | Specimen/culture/ voucher/strain | Locality | References |
|---|---|---|---|---|
| *G. adspersum* | KF605651 | JV 1106/7 | USA | [41] |
| *G. adspersum* | MG706204 | ACAM A113 | Greece | Genbank |
| *G. adspersum* | MN945139 | 341 | Spain | Genbank |
| *G. adspersum* | JN588580 | Ga-1 | Italy | [42] |
| *G. applanatum* | KU219987 | Dai 8924 | China | [43] |
| *G. applanatum* | MF161295 | BHI-F570a | USA | [44] |
| *G. applanatum* | JX501311 | BL26 | France | [45] |
| *G. applanatum* | JQ520162 | IUM 3985 | Netherlands | [46] |
| *G. applanatum* | KY364256 | SFC20141001−25 | Korea | [41] |
| *G. australe* | MH571686 | CMW47785 | South Africa | [47] |
| *G. australe* | KU569545 | RP57 | Colombia | [48] |
| *G. australe* | MF436676 | HUEFS:DHCR417 | Brazil | [49] |
| *G. ellipsoideum* Hapuar., T.C. Wen & K.D. Hyde, sp. nov. (holotype) | NR 160617 | GACP 14080966 | Hainan, China | [25] |
| *G. ellipsoideum* | MH106867 | JFL 14080966 | Hainan, China | [25] |
| *G. ellipsoideum* | MH106886 | JFL14081228 | Hainan, China | [25] |
| *G. ellipsoideum* | MZ354971 | Dai 20544 | China | [50] |
| ***G. ellipsoideum*** | **OQ780769** | **VNES22015-A1** | **Vietnam** | **This study** |
| *G. gibbosum* | AY593856 | AS5.624 type 3 | China | [51] |
| *G. gibbosum* | MK345432 | GZ14070501 | Thailand | [52] |
| *G. gibbosum* | MK280717 | Pvc62 | Taiwan, China | Genbank |
| *G. gibbosum* | KY364264 | SFC20140702−12 | Korea | [41] |
| *G. lobatum* | KF605675 | JV 0409/13J | USA | [52] |
| *G. lobatum* | KF605676 | JV 0409/10J | USA | [52] |
| *G. lobatum* | JQ514104 | URM83328 | Brazil | Genbank |
| *G. philippii* | MG279189 | Cui 14444 | China | [53] |
| *G. philippii* | AJ536662 | E7098 | Malaysia | [54] |
| *G. philippii* | MN401411 | MFLU 19−2223 | Thailand | [26] |
| *G. philippii* | ON202931 | SL1649 | Singapore | Genbank |
| *Tomophagus colossus* | KJ143923 | TC-02 | Vietnam | [55] |

ITS rDNA BLAST sequences via the MAFFT v7.487 [36], to align the sequences obtained. The alignments were edited and manually corrected using AliView. The best fit model was K2+G which was determined by jModelTest 2.1.10 based on the Corrected Akaike Information Criterion (AICc) [37]. Maximum Likelihood (ML) analyses were performed with PhyML 3.0 under K2+G model with 1000 rapid bootstrap replicates. For Bayesian analysis with the same model and posterior probabilities (PP) were determined by Markov Chain Monte Carlo (MCMC) using MrBayes v.3.2.7a [38]. Three different runs with 5000 generations and four chains were executed until the split deviation frequency value < 0.01 and sampled every 100th generation. The initial 25% of sample trees were discarded as burn-in. The MCMC runs were checked reaching convergence with all ESS values above 200 by using Tracer v.1.7.2 [39]. Maximum Likelihood analysis (ML) with 1000 ultrafast bootstrap replicates and Baysian analysis were performed on CIPRES Science Gateway V 3.3 [40]. The ML bootstrap values (BS) ≥ 70% and PP ≥ 0.95 were presented on topologies from ML analyses, respectively. *Tomophagus colossus* TC-02 was selected as the outgroup in both ML and Bayesian analysis. Phylogenetic trees were visualized and modified by FigTree v.1.4.4.

## Liquid-liquid extraction fractions

The 70% ethanol extract of basidiocarps from *Ganoderma ellipsoideum* VNES22015-A1 was prepared according to the methodology described in our previous study [56]. Fruiting bodies were dried at 50 °C to constant weight and ground into powder. A 100 g sample was extracted twice with 1500 mL of 70% ethanol under magnetic stirring (120 rpm, 72 h). The combined extracts were filtered (Whatman No. 4) and concentrated to dryness at 40 °C using a rotary evaporator (IKA RV 10 digital V-C). After that, ten grams of 70% ethanol extract was redissolved in 30 mL 70°C distilled water, cooled down, and then applied to liquid-liquid extraction with *n*-hexane and ethyl acetate, respectively (20 mL each time, five times). The *n*-hexane, ethyl acetate, and aqueous fractions were evaporated at 40°C to obtain dried extractive fractions. The extraction yield was calculated by comparing the dry basidiome biomass and extractive ethanol before extraction to the dry weight after evaporation.

## Cytotoxicity assay

MFC-7 cells were cultured in EMEM with L-glutamine (2 mM), amphotericin B (0.025 μmL), penicillin G (100 UI/mL), streptomycin (100 μg/mL), 10% (*v/v*) FBS in 95% air/5% $CO_2$ at 37°C. Cytotoxic activity was determined by using the sulforhodamine B (SRB) assay with a slight modification [57]. MFC-7 cells ($10^4$ cells/well) were incubated in 96 well plates in the presence of various concentrations of extracts (0, 50, 100, 200, 300, 400 μg/mL) for 48h. 0.25% (*v/v*) DMSO was the negative control and camptothecin (0.5 μg/mL) was the positive control. Then, the total protein from MFC-7 cells was mixed with 25 μL cold 50% solution (*wt/vol*) of trichloroacetic acid in each well, incubated the plates at 4°C for 1h, and then stained with 0.2% SRB solution. Results were measured at 492 nm by using an ELISA REAGEN™ microplate reader (San Diego, CA, USA).

## Protein and ligand preparation

The crystal structures of human placental aromatase (HPA – PDB ID: 3S7S), maternal embryonic leucine zipper kinase (MELK – PDB ID: 5IH9), human protein kinase CK2alpha (CK2a – PDB ID: 5OWH), and NUDIX hydroxylase (NUDT5 – PDB ID: 5NWH) are available on RSCB Protein Data Bank (http://rcsb.org) and were downloaded. All non-standard residues were removed using UCSF Chimera 1.17.3. Polar hydrogen atoms and Gasteiger charges were added to the proteins by the "Dock Prep" tool of UCSF Chimera [58]. The SMILES strings of secondary metabolites from *Ganoderma ellipsoideum* were generated by ChemDraw, along with the PubChem CID of the reference compound of tamoxifen were pasted to the "Build Structure" tool of UCSF Chimera for ligands formation. Ligands preparation was done by "Dock Prep" tool of UCSF Chimera where hydrogen atoms were added and Gasteiger charges were assigned to the ligands. The

energy minimisation of the ligands was executed by the "Minimize Structure" tool of UCSF Chimera. The prepared proteins and ligands were saved in PDB format.

## Molecular docking

In this study, AutoDock Vina 1.2.5 [59], which had been integrated into UCSF Chimera, was used for the molecular docking process. PrankWeb was employed to investigate the binding active sites of the proteins. These binding pockets were ranked and chosen based on the criteria of highest score and probability. To cover the binding active sites, the grids were centered at the area that includes all the residues pointed out by Prankweb. The best conformers were searched based on the Broyden-Fletcher-Goldfarb-Shanno algorithm. For each ligand, the number of conformers was set at a maximum of 10 during the molecular docking process. Default parameters of AutoDock Vina were selected for the docking performance. After the docking process, the conformers were ranked according to their binding energy with the proteins, in which the selection was done on the least binding energy among all generated conformers. All the AutoDock Vina docking performances were run under Windows 10 Pro operating system, with the processor of 2.53 GHz Intel Core i5 [60].

## ADMET predictions

All studied compounds were screened based on "Lipinski's rule of five" [61]. SwissADME web tool was used to calculate data relating to the pharmacokinetics of these compounds. In the case of acute oral toxicity prediction, the algorithm of the DL-AOT Prediction Server was applied.

## Statistical analysis

Quantitative data were obtained from triplicate experiments and the analysis was performed as the mean ± standard deviation. The comparisons between multiple groups were calculated with the one-way ANOVA followed by the Tukey post hoc test in the R software.

## Results and discussion

### Morphology identification

*Habitat:* saprophyte on the stump of hardwood.

   *Basidiomes:* Annual or perennial, stemless, and flabelliform (fan-shaped) when viewed from above (Fig 1a). The pileus features concentric sulcate zones and scattered rays, with the edge being thick, wrinkled, slightly curved inward, and non-laccate. The pileus measures 13–15 × 11–12 cm in diameter and is 0.6–2.4 cm thick. It is imbricated, uneven, broadly attached when mature, and thickened at the base. The pileus surface is non-laccate, silky, and slippery when fresh, but becomes furrowed with sulcate to undulating patterns as it ages. The surface is somewhat spathulate to uneven, incised, compact, hard, and woody when older, often covered with a tough crust (0.2–0.5 mm) that is usually dull and faded in appearance as it matures or ages. In older specimens, the crust may become lined or cracked. The pileus color is homogeneously brownish-orange (5C5) toward the center, dark brown (6F4) at the base, and white at the margin. The color usually darkens upon handling. The lower surface is white (4A1) to greyish-white (4B1), with numerous pores for sporulation (Fig 1b). The context tissue is 0.3–0.9 cm thick, dry, and not completely homogeneous, with continuous bands of resin-like deposits that are brown (6E8) (Fig 1c, d).

   *Hymenophore:* Hymenophore is dark brown (7F6). The tube layers are 0.2–1.5 cm in length (Fig 1c, d). The margin is wavy, blunt, slippery when wet, thinner at the base, and softer than the center, with a white color. There are 3–4 pores per mm. When fresh, the pores are angular, subcircular to circular (Fig 1e, f). The pileipellis is a hymeniderm, ranging from brownish-orange (5C4) to dark brown (7F4), formed by branched hyphal endings that are dextrinoid (Fig 1g). The pore surface is white (4A1) to yellowish-white (4D2) when fresh and becomes discolored upon touch, turning brownish-yellow (6C8) when dry.

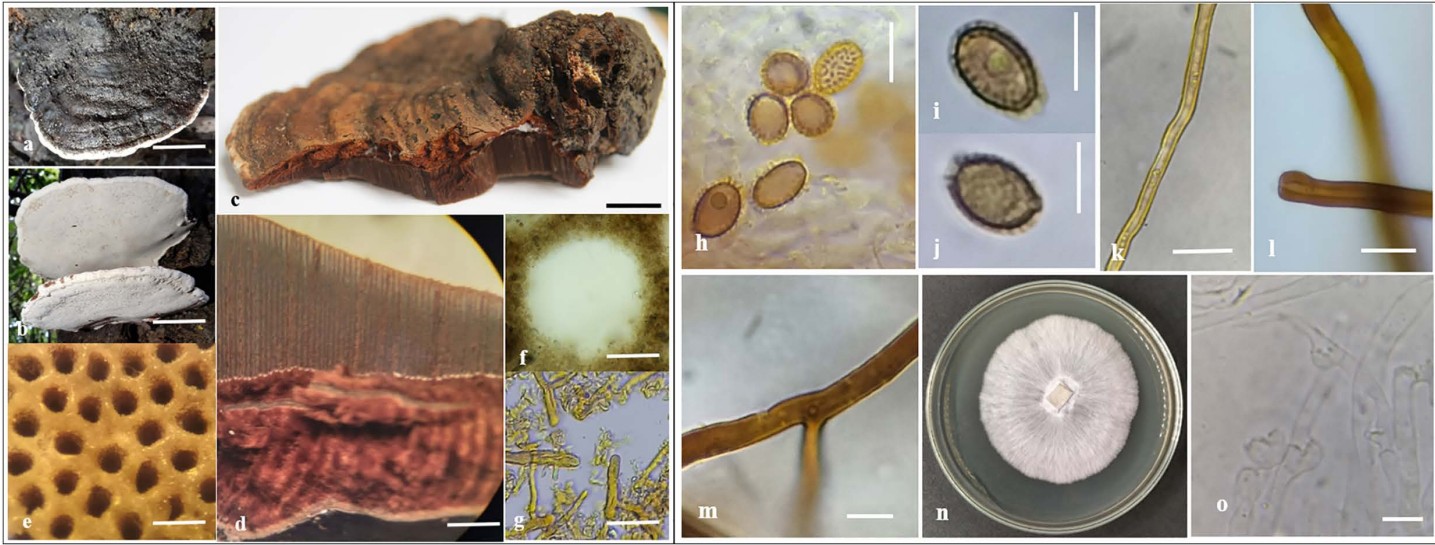

**Fig 1. Morphology of *Ganoderma ellipsoideum* (VNES22015-A1). a.** Basidiome (upper surface); **b.** Basidiome (lower surface). **c.** Cut surface; **d.** Context and hymephore (4×); **e.** Pores in lower surface (4×); **f.** A pore (40×); g. section of pileipellis (100×); **h.** Basidiospores with short and distinct echinulae (100×); i, **j.** Basidiospores with truncate apex and apical germ pore (100×); **k.** Generative hyphae (100×); **l.** Skeletal hyphae (100×); **m.** Binding hyphae (100×); **n.** Mycelia on PDA medium (6 days old) at 25±2ºC; **o.** Hyphae and camp connection on PDA medium. Scale bars: a – c: 4 cm; d, e: 50 mm; f: 50 μm, h-j: 5 μm; g: 35 μm; k – n: 3 μm.

*Basidiospores:* Basidiospores are mainly ovoid, truncated, and have obvious double walls (Fig 1h-j). The size of basidiospores with myxosporium (n = 50) ranges from (5.2-)7.1–7.6(−9.3) × (3.8-)5.0–5.3(−6.3) μm, with an average basidiospore length ($L_{avg}$) of 7.3±1.10 μm, an average basidiospore width ($W_{avg}$) of 5.1±0.64 μm, and an average Q-value ($Q_{avg}$) of 1.4±0.14. For basidiospores without myxosporium (n = 50), the size ranges from (3.9-)5.7–6.2(−8.0) × (3.0-)3.9–4.2(−5.3) μm, with an average basidiospore length ($L_{avg}$) of 5.9±0.98 μm, an average basidiospore width ($W_{avg}$) of 4.0±0.57 μm, and an average Q-value ($Q_{avg}$) of 1.5±0.17. Basidiospores have germ pores, and their inner wall is yellowish-brown (5D8) to greyish-brown (5E3) in 5% KOH, overlaid by a hyaline layer.

*Hyphal structure*: The hyphal system is trimitic, with clamp connections (Fig 1k-m). Generative hyphae are 1.2–3.3 μm wide (n = 20), with walls of varying thickness, yellow to light brown in 5% KOH, and unbranched. Skeletal hyphae are 4.0–6.0 μm wide (n = 20), light brown to brown in 5% KOH, and usually thick-walled. Binding hyphae are 2.5–5.5 μm wide (n = 20), light brown to brown in 5% KOH, thick-walled, with many branches. The generative and skeletal hyphae are usually intertwined. Cross-sections of pores, with surrounding sporulation in 5% KOH, show spores and undeveloped mycelium, and the pores are round, measuring 6–7 μm in diameter.

*Colonies* grow on PDA medium at 25±2°C. The mycelium is uniformly white, resembling absorbent cotton, with radial growth that is thick in the center and thinner at the edges after 6 days (Fig 1n). The filamentous system proliferates on PDA medium and is fully grown after 8 days in Petri dishes (90 mm diameter). No chlamydospore was detected on the medium after 8 days of growth (Fig 1o).

*G. ellipsoideum* was found for the first time in Jiangfenling Mountain, Hainan, China China (25). This species belongs to G. *applanatum-australe* complex (*G. tornatum, G. adspersum, G. lobatum, G. philippii, G. pseudoferreum, G. gibbosum*) with the same characteristics: non-laccate pileus the absent of chamydospores in culture [62]. It is separated from other *Ganoderma* species by elongate to ellipsoid basidiospores, the length of spores is smaller than 8 μm, the context is thick and brown (about 4–5 mm), and long tubes.

## Phylogenetic analysis

The phylogenetic tree was based on ITS rDNA sequences that are shown in Fig 2. These sequences belong to the *G. applanatum-australe* complex and were grouped into six clades. Twenty-seven sequences *G. applanatum-australe* complex were obtained from GenBank (from samples collected in Brazil, Colombia, South Africa, USA, France, Greece, Italy, Spain, Korea, China, Malaysia, Singapore, Thailand, and Vietnam) with a holotype (*G. ellipsoideum* GACP140966). *Tomophagus colossus* (Fr.) Murill 1905 (TC-02, Vietnam) is the outgroup of the taxonomic tree. Tree topologies resulting from the Maximum Likelihood (ML) analysis were similar to the Bayesian analysis. The best-scoring ML tree and Bayesian analysis are shown in Fig 2. The results showed that specimens collected from Vietnam clustered significantly, forming a monophyletic group with the *G. ellipsoideum* taxon, including the holotype (GACP 1480966) and other sequences from China (JFL 14080966, JFL 14081228, and Dai 20544), with high ML/PP support values (90% and 1.00, respectively).

From the phylogenetic tree in Fig 2, it is evident that *Ganoderma ellipsoideum* is more closely related to *G. gibbosum* within the *G. applanatum-australe* complex. While *G. philippii* and *G. applanatum* form separate clades, *G. ellipsoideum* clusters with *G. gibbosum, G. lobatum, G. australe,* and *G. adspersum* in a distinct clade. This clade is more closely related to the one containing strains of *G. applanatum* than to the clade containing strains of *G. philippii*. This phylogenetic evidence aligns with the morphological characteristics observed among species in the *G. applanatum-australe* complex.

## Yield of extraction

In this study, hydroethanolic solvent (70% ethanol: 30% water) was used to extract compounds because this solvent can effectively break down fungal cells, and also created intermolecular hydrogen bonds with polar groups, and the desired

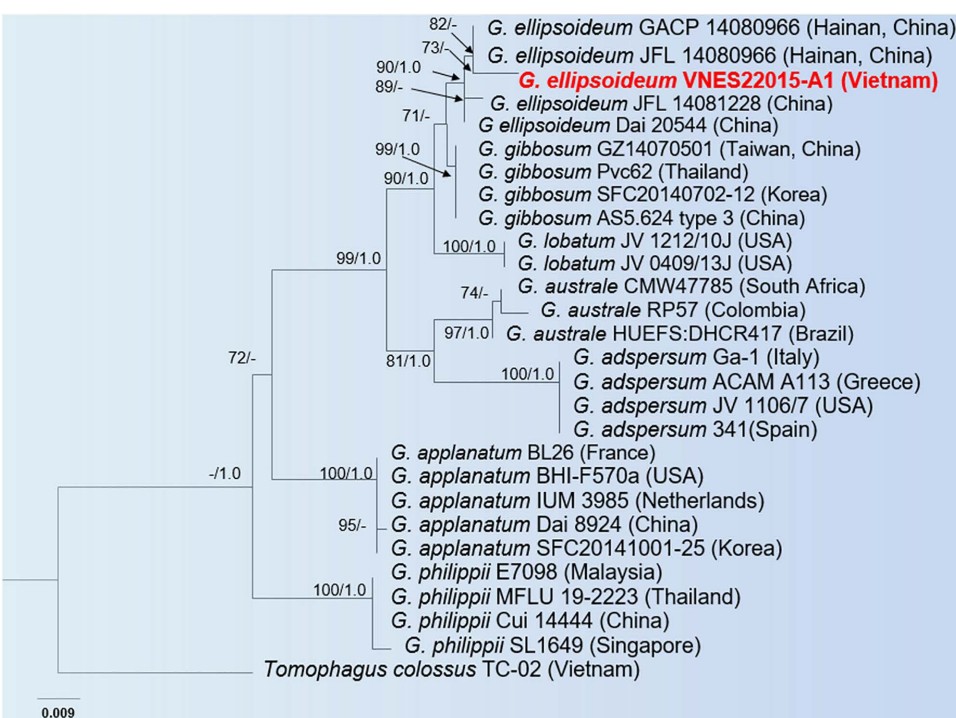

**Fig 2. Phylogeny of *G. applanatum-australe* complex based on the data from ITS-5.8S rDNA sequences.** Bootstrap values were obtained from Maximum Likelihood above 70% and posterior probabilities ≥ 0.95 that were given above branches. The tree is rooted in *Tomophagus colossus* TC-02 (Vietnam). New species were collected in Vietnam for this study in bold red.

extractive compounds in the matrix. This solvent can extract compounds with strong, moderate, and weak polarity. Then, the filtered liquid extracts were liquid-liquid extracted with n-hexane and ethyl acetate solvents having increased polarities. Extraction yields are shown in Table 2.

The result indicated that the 70% ethanol extraction yield was 4.29±0.27%. Compared to our previous study using 70% ethanol for extraction, the extraction of *G. ellipsoideum* was higher than *G. multiplicatum* (2.40±0.06%) and lower than *G. sinense* (5.68%±0.27%) [56]. The extract of *G. ellipsoideum* VNES22015-A1 was higher than ethanol 95% of *G. lucidum*, 3.05% [63] and methanolic extract of *G. tsugae*, 3.97% [64]. Using organic solvents for isolating compounds that are raising dielectric constants, given the information in Table 2, the aqueous fraction was obtained as the highest percentage of the three fractions, followed by the ethyl acetate fraction, and the last n-hexane fraction at 56.33±1.21, 38.17±0.73, 3.25±0.50%, respectively. The dielectric constants of organic solvents rose which shows a similar pattern as the weight proportion increased. That illustrates that the compounds of basidiocarps *G. ellipsoideum* are mostly polar and hydrophilic.

## Cytotoxicity activity result

The properties of proliferation of fruiting bodies *G. ellipsoideum* ethanol and solvent-partitioned fractions, using Sulforhodamine B colorimetric assay, are present in Table 3. The results showed that at 50 µg/mL of the 70% mushroom extraction, the n-hexane GEFH fraction and the aqueous fraction had no toxic activity, while ethyl acetate fraction at 50 µg/mL described the inhibition ability to 12.22±0.55% of MFC-7 cells. Moreover, when the concentration of

**Table 2. Yield of extraction from fruiting bodies *G. ellipsoideum* VNES22015-A1 (*wt/wt*).**

| Solvent extraction | 70% ethanol extract | *n*-hexane fraction | Ethyl acetate fraction | Aqueous fraction |
|---|---|---|---|---|
| Percentage (%) | 4.29±0.27[a] | 3.25±0.50[b] | 38.17±0.73[b] | 56.33±1.21[b] |

[a]Yield of 70% ethanol crude extract from 100 g dried ground basidiocarps. [b]Yield of fractions from 10 g 70% ethanol crude extract. The R software was used to obtain means and standard deviations of three different experiments.

**Table 3. Cytotoxic activity results.**

| Samples | Concentration (µg/mL) | MCF-7 cells inhibition (%) | Samples | Concentration (µg/mL) | MCF-7 cells inhibition (%) |
|---|---|---|---|---|---|
| GEE | 50 | – | GEFH | 50 | – |
| | 100 | 6.29±1.07 | | 100 | 3.50±0.52 |
| | 200 | 19.93±0.55 | | 200 | 17.49±0.50 |
| | 295.87±7.45 | 50 | | 300 | 38.50±0.70 |
| | 300 | 50.79±0.61 | | 356.46±6.78 | 50 |
| | 400 | 91.43±0.87 | | 400 | 62.03±1.13 |
| GEFE | 50 | 12.22±0.55 | GEFA | 50 | – |
| | 100 | 27.34±0.96 | | 100 | 8.47±0.56 |
| | 150 | 41.07±1.23 | | 200 | 21.20±0.71 |
| | 171.13±3.76 | 50 | | 300 | 42.45±0.81 |
| | 200 | 59.06±0.81 | | 325.86±6.51 | 50 |
| | 300 | 99.21±1.12 | | 400 | 95.49±0.87 |
| DMSO (0.25%) | | – | Camp | 0.5 | 51.27 ±1.77 |

GEE: ethanol extract; GEFE: ethyl acetate fraction; GEFH: *n* – hexane fraction; aqueous fraction. Negative control: 0.25% DMSO (Dimethyl sulfoxide); Camp: camptothecin 0.5 µg/mL (positive control). The results represent the mean±SD of three experiments. (-) Not detected.

fractions was increased, the inhibition activity to cancer cells of extraction was higher, in particular, the ethyl acetate fraction had a more effective inhibitory ability than 70% ethanol, n-hexane, and aqueous fractions. Therefore, in Table 3, $IC_{50}$ of the ethyl acetate extract was lower than that of the 70% ethanol extract, the n-hexane, and water fractions, $171.13 \pm 3.76$ µg/mL compared to $295.87 \pm 7.45$ µg/mL of GEE, $356.46 \pm 6.78$ µg/mL of GEFH, and $325.86 \pm 6.51$ µg/mL of GEFA, respectively. This $IC_{50}$ value indicated that *G. elipsoideum* VNES22015-A1 has a higher cytotoxicity effect on breast cancer cells MCF-7 than hot water extracts of *G. lucidum* and *Amauroderma rude* in a previous study of Jiao and coworkers (2013) with above 90% of inhibited cells at 600 µg/mL [14]. In our study, total 70% ethanol extract at 400 µg/mL inhibited $91.43 \pm 0.87\%$ of cancer cells, GEFA fraction $95.49 \pm 0.87\%$ and GEFE fraction $99.21 \pm 1.12\%$ of cancer cells.

The previous reports have indicated the main compounds of *Ganodermataceae* including triterpenoid, steroid, and polysaccharides which were extracted in solvents with different polarities. Ethanol extract of *G. lucidum* at 500 µg/mL inhibited 70% of MCF-7 cancer cells through apoptosis induction [65]. Studies on *G. resinaceum* mycelium of ergosterol peroxide and ganoderic acid AM1 isolated from ethyl acetate extract showed inhibitory effects on MCF-7 cells [23]. Another report showed the extract of *G. lucidum* (GLE) increased reactive oxygen species (ROS) level and caused pyrotopic cell death [66]. GLE also activated caspase 3 to cleave gasdermine E, thereby generating pores on the breast cancer cell membrane. Studies on polysaccharides of *G. applanatum* indicated that at 500 µg/mL, 50.2% of MCF-7 cells were inhibited by apoptosis and autophagy via the MAPK signaling pathway [15].

### Interaction and binding affinity of compounds in the active site of HPA enzyme

In the next study, molecula docking simulation (S1 Fig), and the free binding energies of the ten compounds as ganoellipsic acid A-C, applanoxidic acid C, applanoxidic acid G, 20-hydroxy-3,12,15,23-tetraoxolanosta-7,9(11),16-trien-26-oic acid, applanoxidic acid A, applanoxidic acid E, applanoxidic acid F, and gibbosicolid D (**1–10**), which were isolated from *G. ellipsoideum* [27], were calculated by AutoDock Vina toward HPA target (Table 4).

**Table 4. Docking results toward HPA.**

| Ligand | Compound | Docking score (kcal/mol) | Hydrogen bond | Hydrophobic interaction |
|---|---|---|---|---|
| 1 | Ganoellipsic acid A | −9.3 | | Ile133, Leu152, Ala306, Met311, Phe430, Ala443, Met447 |
| 2 | Ganoellipsic acid B | −8.7 | Thr310 | Ile133, Phe148, Ala306, Ala307, Met311, Met446 |
| 3 | Ganoellipsic acid C | −9.3 | Ala307, Val373, Met374 | Ile133, Ala306, Ala307, Cys437 |
| 4 | Applanoxidic acid C | −9.6 | Ala438 | Leu152, Ala306, Met311, Phe430, Ala443, Met447 |
| 5 | Applanoxidic acid G | −9.3 | Arg375, Gly431, Gly436 | Ile133, Phe134 |
| 6 | 20-Hydroxy-3,12,15,23-tetraoxolanosta-7,9(11),16-trien-26-oic acid | −9.2 | Ala438 | Ile133, Leu152, Ala306, Met311, Phe430, Ala443, Met447 |
| 7 | Applanoxidic acid A | −9.4 | Met303, Ala306, Ala307 | Ile133, Cys437 |
| 8 | Applanoxidic acid E | −9.4 | Arg115, Arg375, Gly431, Arg435 | Ile133, Val370 |
| 9 | Applanoxidic acid F | −9.5 | Ser199, Met303, Ala306, Ala307, Cys437, Ala438 | Ile133 |
| 10 | Gibbosicolid D | −8.8 | Gln428, Tyr441 | Tyr424, Tyr441, Met444 |
| | Tamoxifen[a] | −8.2 | | Arg115, Ile132, Ile133, Met303, Ala306, Arg435, Cys437, Ala438 |

[a]Positive control

The best docked poses of the ligands are displayed in Fig 3, pointing out their interactions with the binding active site of the protein. Basically, all studied compounds showed the potential of strong inhibitors since these ligands formed several hydrogen bonds as well as hydrophobic interactions with residues in the active binding site of the HPA enzyme. Notably, the free binding energies of all *Ganoderma ellipsoideum*'s components towards the HPA enzyme were ranked from −9.6 to −8.8 kcal/mol. In which, applanoxidic acid C and applanoxidic acid F were suggested as strong inhibitors of the HPA enzyme with free binding energy values of −9.6 and −9.5 kcal/mol, respectively, while that of tamoxifen was −8.2 kcal/mol, meaning these ligands were all predicted to have stronger affinity towards the target enzyme than the reference ligand of tamoxifen.

### Interaction and binding affinity of compounds in the active site of MELK enzyme

The free binding energies of the ligands (**1**–**10**) were also calculated by AutoDock Vina, providing the basis for assessing the affinity of the ligands toward MELK target (Table 5).

The docking simulations of MELK protein with all studied ligands revealed the potential of these compounds as MELK inhibitors since several interactions formed between the ligands and residues in the receptor's active site. S2 Fig in Supporting information exhibits the best docked poses of the ligands, highlighting their binding interactions with the receptor. Interestingly, there was a similarity in the value of free binding energy of ganoellipsic acid A, and tamoxifen toward MELK receptor, which was −7.9 kcal/mol. The remaining studied ligands all had stronger affinity towards the target enzyme with the free binding energy ranked from −9.6 to −8.0 kcal/mol, in which applanoxidic acid F was predicted to show the strongest affinity toward MELK protein with the binding free energy of −9.6 kcal/mol. This ligand formed several interactions with the target, including six hydrogen bonds with Thr19, Lys40, Arg53, Glu57, Cys89, and Gly152 along with four hydrophobic interactions with Phe22, Ala23, Met42, and Ile54 (Fig 4).

### Interaction and binding affinity of compounds in the active site of CK2a enzyme

The free binding energies of the ligands were also calculated by AutoDock Vina, providing the basis for assessing the affinity of the ligands toward CK2a target (Table 6).

Fig 5 displays the best docked poses of studied ligands, highlighting their binding interactions with the CK2a receptor. While the free binding energy of the reference ligand was −8.4 kcal/mol, the remaining studied ligands showed strong affinity towards CK2a inhibitors since their values ranked from −9.5 to −8.8 kcal/mol (S3 Fig). Specifically, applanoxidic

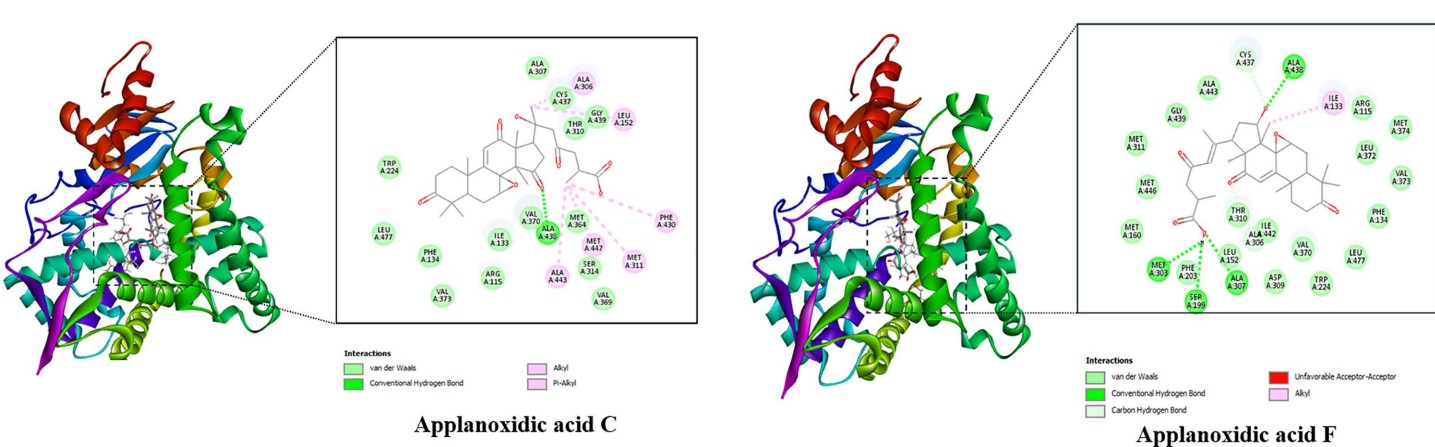

**Fig 3. Top two compounds and HPA interactions.**

**Table 5. Docking results toward MELK.**

| Ligand | Compound | Docking score (kcal/mol) | Hydrogen bond | Hydrophobic interaction |
|---|---|---|---|---|
| 1 | Ganoellipsic acid A | −7.9 | Gly20, Glu93, Asp150 | Phe95 |
| 2 | Ganoellipsic acid B | −8.8 | Phe22, Lys40, Ile54, Cys89, Asp150 | Phe22, Ala23, Val25, Lys40, Met42, Ile54, Ile149 |
| 3 | Ganoellipsic acid C | −9.2 | Phe22, Lys40, Arg53, Ile54, Glu57, Cys89, Gly152 | Phe22, Ala23, Met42, Ile54, Ile149 |
| 4 | Applanoxidic acid C | −8.0 | Glu93 | Phe95 |
| 5 | Applanoxidic acid G | −8.3 | Thr19, Gly20, Glu93 | Phe95 |
| 6 | 20-Hydroxy-3,12,15,23-tetraoxolanosta-7,9(11),16-trien-26-oic acid | −8.3 | | Ile17, Leu139 |
| 7 | Applanoxidic acid A | −8.3 | Lys40, Lys134 | |
| 8 | Applanoxidic acid E | −8.4 | Glu93, Lys134 | Ala23, Val25 |
| 9 | Applanoxidic acid F | −9.6 | Thr19, Lys40, Arg53, Glu57, Cys89, Gly152 | Phe22, Ala23, Met42, Ile54 |
| 10 | Gibbosicolid D | −9.1 | Lys40 | Ala23, Val25 |
| | Tamoxifen[a] | −7.9 | Cys89 | Val25, Ala38, Lys40, Leu86, Leu139, Ile149 |

[a]Positive control.

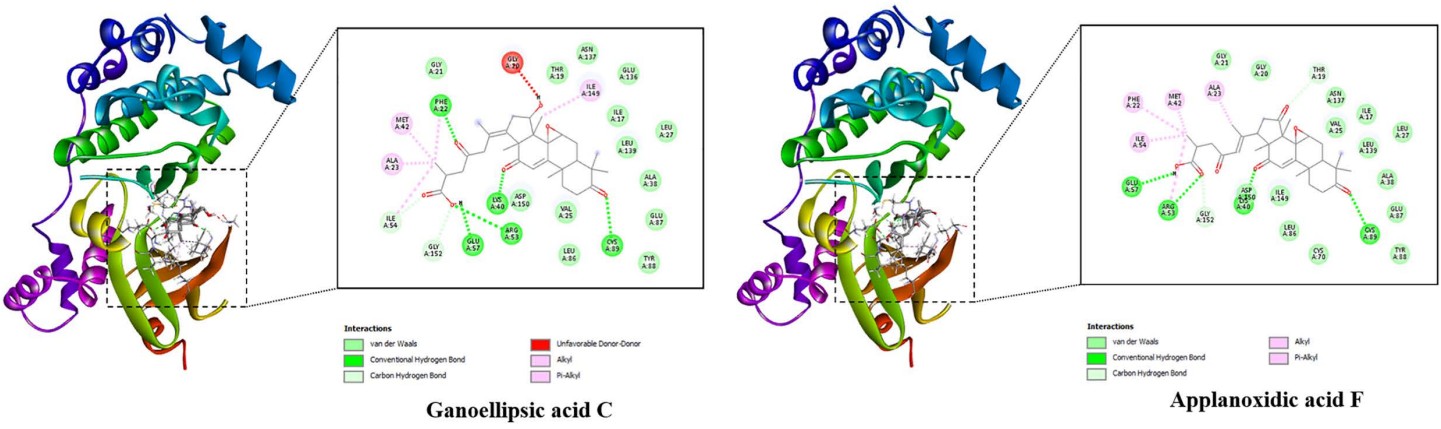

**Fig 4. Top two compounds and MELK interactions.**

acid F demonstrated the potential of CK2a protein inhibitor as the value of free binding energy was −9.5 kcal/mol. This ligand formed hydrophobic interactions with Val53, Lys68, Ile95, Phe113, His160, and Ile174.

## Interaction and binding affinity of compounds in the active site of NUDT5 enzyme

The free binding energies of the ligands were also calculated by AutoDock Vina, providing the basis for assessing the affinity of the ligands toward NUDT5 target (Table 7).

The best docked poses of the ligands are displayed in Fig 6, pointing out their interactions with the binding active site of the protein. Basically, all studied compounds showed the potential of strong inhibitors since these ligands formed several hydrogen bonds as well as hydrophobic interactions with residues in the active binding site of NUDT5 enzyme (Supporting

**Table 6. Docking results toward CK2a.**

| Ligand | Compound | Docking score (kcal/mol) | Hydrogen bond | Hydrophobic interaction |
|---|---|---|---|---|
| 1 | Ganoellipsic acid A | −9.2 | Gly48, Lys68 | Val53 |
| 2 | Ganoellipsic acid B | −8.8 | Lys68, Val116, Asn117 | Val53, Lys68, Asp175 |
| 3 | Ganoellipsic acid C | −9.3 | | Val53, Val66, Lys68, Ile95, Phe113, His160, Ile174 |
| 4 | Applanoxidic acid C | −9.4 | Gly48, Lys68, Val116, Asn117 | Val53, Lys68 |
| 5 | Applanoxidic acid G | −9.0 | Lys68, Asp175 | Val53, Lys68, Ile95, Phe113, His160, Met163, Ile174 |
| 6 | 20-Hydroxy-3,12,15,23-tetraoxolanosta-7,9(11),16-trien-26-oic acid | −9.0 | Lys68 | Val53, Met163 |
| 7 | Applanoxidic acid A | −9.3 | Asp175 | Val53, Lys68, Ile95, Phe113, His160, Ile174 |
| 8 | Applanoxidic acid E | −8.9 | Lys68, Asp175 | Val53, Ile95, Phe113, Met163, Ile174 |
| 9 | Applanoxidic acid F | −9.5 | Gly48 | Val53, Lys68, Ile95, Phe113, His160, Ile174 |
| 10 | Gibbosicolid D | −8.9 | | Val53, Val66, Ile95, Phe113, His160 |
| | Tamoxifen[a] | −8.4 | Gly48, Lys49 | Val53, Val66, Lys68, Asp153, Met163, Ile174, Asp175 |

[a]Positive control.

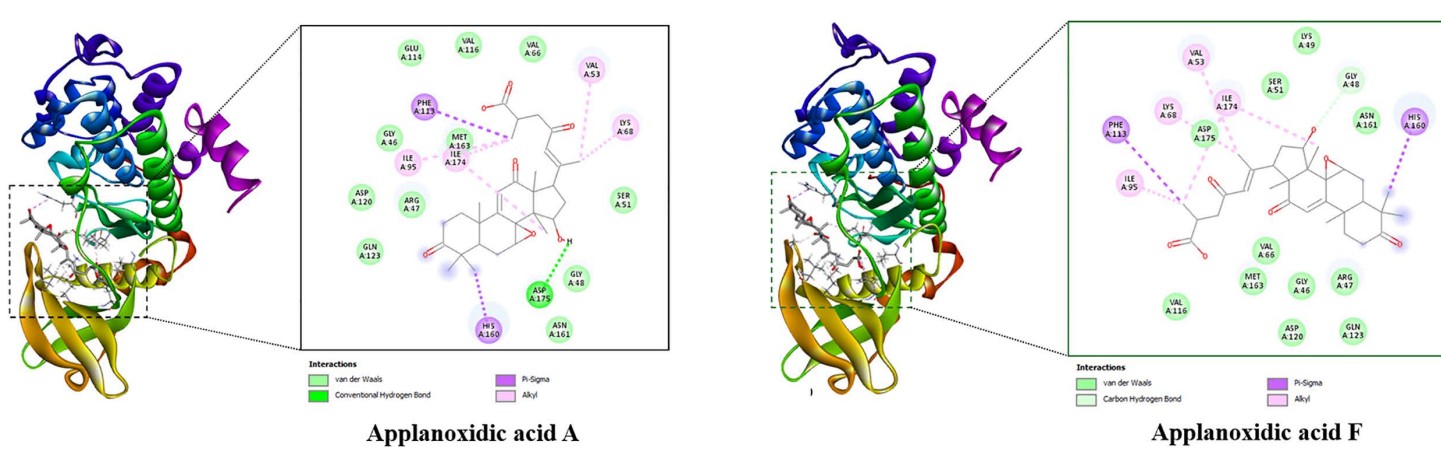

**Applanoxidic acid A**          **Applanoxidic acid F**

**Fig 5. Top two compounds and CK2a interactions.**

information S4 Fig). Notably, the free binding energies of all *Ganoderma ellipsoideum*'s components towards HPA enzyme were ranked from −9.9 to −8.9 kcal/mol. In which, gibbosicolid D was suggested as strong inhibitor of the NUDT5 enzyme with free binding energy of approximately −9.9 kcal/mol, while that of tamoxifen was −8.1 kcal/mol, meaning these ligands were all predicted to have stronger affinity towards the target enzyme than the reference ligand of tamoxifen. Notably, Applanoxidic acid F, as pointed out as a potential inhibitor of anti-breast cancer targets, also showed strong affinity towards NUDT5 enzyme with a binding energy of −9.3 kcal/mol (Fig 6).

**Table 7. Docking results toward NUDT5.**

| Ligand | Compound | Docking score (kcal/mol) | Hydrogen bond | Hydrophobic interaction |
|---|---|---|---|---|
| 1 | Ganoellipsic acid A | −9.0 | Arg44, Arg51, Gly61, Val62, Arg84, Gly165 | Ala96, Met132, Ile141 |
| 2 | Ganoellipsic acid B | −8.9 | Arg51, Gly61, Arg196 | Trp46 |
| 3 | Ganoellipsic acid C | −9.5 | Arg44, Gly61, Val62, Arg84, Gly165 | Ala96, Met132, Ile141 |
| 4 | Applanoxidic acid C | −9.3 | Arg44, Arg51, Gly61, Val62, Arg84, Gly165 | Ala96, Met132, Ile141 |
| 5 | Applanoxidic acid G | −9.0 | Arg44, Arg51, Gly61, Arg84 | Ala96, Met132, Ile141 |
| 6 | 20-Hydroxy-3,12,15,23-tetraoxolanosta-7,9(11),16-trien-26-oic acid | −8.9 | Gly61, Arg84, Gly165 | Trp28, Leu98, Cys139 |
| 7 | Applanoxidic acid A | −9.3 | Arg44, Arg51, Gly61, Val62, Arg84, Gly135, Gly165 | |
| 8 | Applanoxidic acid E | −9.1 | Arg44, Arg51, Arg84, Gly97, Gly135, Gly165 | |
| 9 | Applanoxidic acid F | −9.3 | Arg44, Arg51, Arg84, Gly97, Gly135, Gly165 | |
| 10 | Gibbosicolid D | −9.9 | Arg44, Arg84, Gly135, Gly165 | Ala96, Leu98, Met132 |
| | Tamoxifen[a] | −8.1 | Leu98, Asp153 | Trp28, Trp46, Arg51, Leu98, Asp153 |

[a]Positive control.

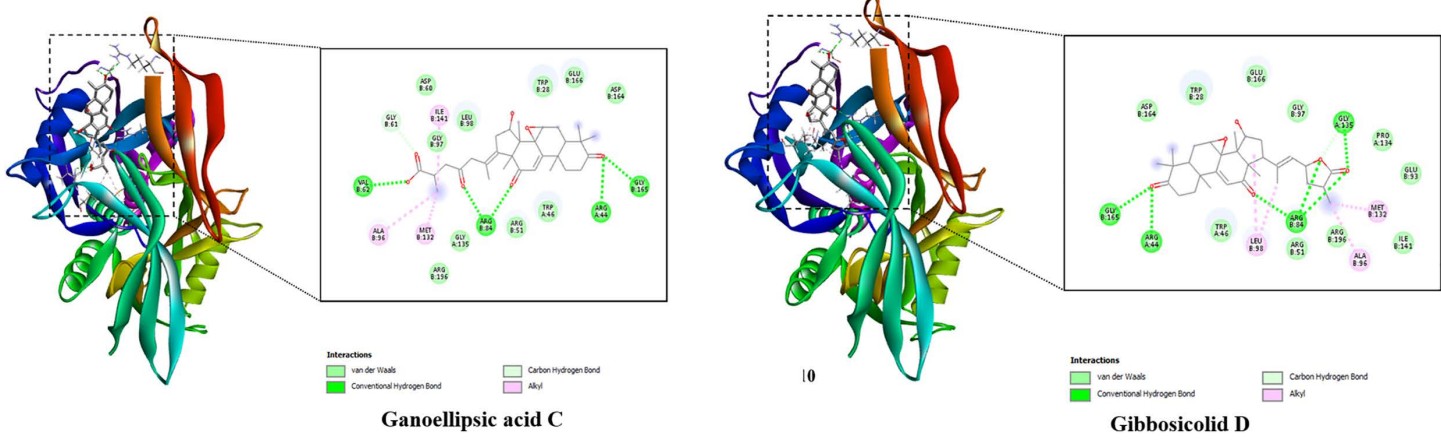

**Fig 6. Top two compounds and NUDT5 interactions.**

## ADMET predictions

The classic "Lipinski's Rule of Five" has traditionally served as a criterion for assessing a compound's druggability. In this study, gibbosicolid D has molecular weight under 500 Daltons, while the remaining *Ganoderma ellipsoideum*'s components (**1–10**) have values over 500 Daltons, meaning they violate the classic rule. All studied compounds also exhibit fewer than 5 hydrogen bond donors, and fewer than 10 hydrogen bond acceptors. Except for tamoxifen with log P value

greater than 5, all remaining compounds satisfy Lipinski's Rule of Five. Additionally, we evaluated the number of rotatable bonds, total polar surface area (TPSA), and aqueous solubility (log S) as physicochemical parameters. To ensure good oral bioavailability and intestinal absorption, the number of rotatable bonds should not exceed 10, and the TPSA value should stay below 140 Å2 [67]. Comprehensive data on these compounds are provided in Table 8, illustrating good physicochemical properties. Furthermore, Table 9 presents *in silico* predictions of the ADME (Absorption, Distribution, Metabolism, and Excretion) properties of the studied compounds. These compounds were predicted not to permeate the blood-brain barrier. Several cytochrome P enzymes play a crucial role in drug biotransformation, including CYP1A2, CYP2C19, CYP2C9, CYP2D6, and CYP3A4. All ten *Ganoderma ellipsoideum*'s components were predicted not to inhibit

**Table 8. Physicochemical properties analyzed with SwissADME.**

| Compound | MW (g/mol) | LogP | nHBD | nHBA | TPSA | MR | Lipinski Violation | LogS | nRotB |
|---|---|---|---|---|---|---|---|---|---|
| Ganoellipsic acid A | 528.63 | 2.51 | 3 | 8 | 141.50 | 139.62 | 1 | −2.87 | 6 |
| Ganoellipsic acid B | 512.63 | 2.90 | 3 | 7 | 128.97 | 140.14 | 1 | −3.31 | 6 |
| Ganoellipsic acid C | 510.62 | 3.25 | 2 | 7 | 121.27 | 137.95 | 1 | −3.92 | 5 |
| Applanoxidic acid C | 526.62 | 2.46 | 2 | 8 | 138.34 | 138.66 | 1 | −2.87 | 6 |
| Applanoxidic acid G | 528.63 | 2.38 | 3 | 8 | 141.50 | 139.62 | 1 | −2.87 | 6 |
| 20-Hydroxy-3,12,15,23-tetraoxolanosta-7,9(11),16-trien-26-oic acid | 510.62 | 3.10 | 2 | 7 | 125.81 | 139.18 | 1 | −3.32 | 6 |
| Applanoxidic acid A | 512.63 | 3.31 | 2 | 7 | 121.27 | 138.42 | 1 | −4.15 | 5 |
| Applanoxidic acid E | 512.63 | 3.33 | 2 | 7 | 121.27 | 138.42 | 1 | −4.15 | 5 |
| Applanoxidic acid F | 510.52 | 3.33 | 1 | 7 | 118.11 | 137.46 | 1 | −3.94 | 5 |
| Gibbosicold D | 496.64 | 3.88 | 1 | 6 | 93.20 | 135.62 | 0 | −4.81 | 2 |
| Tamoxifen | 371.51 | 5.77 | 0 | 2 | 12.47 | 119.72 | 1 | −6.59 | 8 |

MW: molecular weight; log P: log of octanol/water partition coefficient; nHBD: number of hydrogen bond donor(s); nHBA: number of hydrogen bond acceptor(s); TPSA: total polar surface area; MR: molar refractivity; log S: log of solubility; nRotB: number of rotatable bond(s).

**Table 9. ADME predictions computed by SwissADME.**

| Compound | Log Kp (cm/s) | GI Abs | BBB per | Inhibitor Interaction | | | | | |
|---|---|---|---|---|---|---|---|---|---|
| | | | | P-gp substrate | CYP1A2 Inhibitor | CYP2C19 Inhibitor | CYP2C9 Inhibitor | CYP2D6 Inhibitor | CYP3A4 Inhibitor |
| Ganoellipsic acid A | −9.35 | Low | No | No | No | No | No | No | No |
| Ganoellipsic acid B | −8.65 | Low | No | Yes | No | No | No | No | No |
| Ganoellipsic acid C | −8.01 | High | No | Yes | No | No | No | No | No |
| Applanoxidic acid C | −9.33 | Low | No | No | No | No | No | No | No |
| Applanoxidic acid G | −9.35 | Low | No | No | No | No | No | No | No |
| 20-Hydroxy-3,12,15,23-tetraoxolanosta-7,9(11),16-trien-26-oic acid | −8.62 | Low | No | Yes | No | No | No | No | No |
| Applanoxidic acid A | −7.78 | High | No | Yes | No | No | No | No | No |
| Applanoxidic acid E | −7.78 | High | No | Yes | No | No | No | No | No |
| Applanoxidic acid F | −7.99 | High | No | Yes | No | No | No | No | No |
| Gibbosicold D | −7.05 | High | No | Yes | No | No | No | No | No |
| Tamoxifen[a] | −3.50 | Low | No | Yes | No | Yes | No | Yes | No |

[a]Positive control.

CYP1A2, CYP2C19, CYP2C9, CYP2D6, and CYP3A4. The $LD_{50}$ values were calculated using the DL-AOT Prediction Server and found to be between 2.82 and 3.36 (Table 10).

Despite the promising findings, this study has certain limitations that should be acknowledged. First, the cytotoxicity evaluation was limited to a single breast cancer cell line as MCF-7, which may not comprehensively represent the heterogeneity of breast cancer subtypes, particularly triple-negative or HER2-positive variants. Second, although molecular docking results suggested strong binding affinities of triterpenoids from *G. ellipsoideum* to key breast cancer-related proteins, these interactions remain theoretical and require further validation through biochemical or biophysical assays, such as surface plasmon resonance or enzyme inhibition studies. In addition, the active compounds responsible for cytotoxic effects in the ethyl acetate fraction were not isolated and structurally confirmed in this study. Future work should focus on compound isolation, mechanism-based assays such as apoptosis markers, cell cycle arrest, and testing on additional cancer cell lines and *in vivo* models to fully elucidate the therapeutic potential of *G. ellipsoideum*.

## Conclusion

This study provides the first confirmed record of *Ganoderma ellipsoideum* in Vietnam, based on comprehensive morphological characteristics and ITS rDNA sequence analysis. Phylogenetic inference placed the Vietnamese strain within a well-supported monophyletic clade of *G. ellipsoideum* (90% ML/ 1.00 PP), consistent with its non-laccate morphology and affiliation to the *G. applanatum–australe* complex. Bioactivity assays demonstrated that the ethanol extract and its solvent-partitioned fractions exhibit cytotoxic effects against MCF-7 breast cancer cells, with the ethyl acetate fraction showing the strongest inhibitory activity. Furthermore, molecular docking revealed strong binding affinities between major triterpenoids and key oncogenic proteins (HPA, MELK, CK2α, and NUDT5), supporting their potential mechanisms of action. Collectively, these findings not only document *G. ellipsoideum* as a newly identified fungal species in Vietnam but also underscore its promise as a source of bioactive compounds for breast cancer therapy.

Although *G. ellipsoideum* was first reported in China in 2018 and has since been cultivated in Thailand, our study is the first to document its discovery and distribution in Vietnam, specifically within the Ea So Nature Reserve of Dak Lak Province. Due to the limited number of natural fruiting bodies collected, compound isolation from Vietnamese samples was not yet possible. Instead, we prepared 70% ethanol extracts and fractionated extracts (*n*-hexane, ethyl acetate, and water) to evaluate their cytotoxicity against MCF-7 human breast cancer cells. To our knowledge, this

**Table 10. Toxicity predicted by DL-AOT Prediction Server.**

| Compound | $LD_{50}$ (mg/kg) | Toxicity |
|---|---|---|
| Ganoellipsic acid A | 2.83 | Caution |
| Ganoellipsic acid B | 3.27 | Caution |
| Ganoellipsic acid C | 3.08 | Caution |
| Applanoxidic acid C | 2.93 | Warning |
| Applanoxidic acid G | 2.92 | Warning |
| 20-Hydroxy-3,12,15,23-tetraoxolanosta-7,9(11),16-trien-26-oic acid | 3.08 | Caution |
| Applanoxidic acid A | 2.82 | Caution |
| Applanoxidic acid E | 3.31 | Caution |
| Applanoxidic acid F | 3.10 | Caution |
| Gibbosicolid D | 3.36 | Caution |
| Tamoxifen | 3.24 | Caution |

Based on these predicted results, applanoxidic acid C and applanoxidic acid G fall into the "Warning" group, while the remaining compounds were classified as "Caution" ones.

is also the first study worldwide to assess the *in vitro* cytotoxic effects of this mushroom species on human breast cancer cells. Notably, we further investigated the inhibitory potential of its triterpenoids using *in silico* models targeting four key proteins associated with breast cancer (HPA, MELK, CK2α, and NUDT5). These findings provide a valuable foundation for future research, including evaluations across additional breast cancer cell lines, compound isolation, *in vivo* experiments, and gene expression studies (e.g., Western blot) to comprehensively assess the anti-breast cancer potential of *G. ellipsoideum*.

## Supporting information

**S1 Fig. Compounds and HPA interactions.**
(PNG)

**S2 Fig. Compounds and MELK interactions.**
(PNG)

**S3 Fig. Compounds and CK2a interactions.**
(PNG)

**S4 Fig. Compounds and NUDT5 interactions.**
(PNG)

## Acknowledgments

The authors would like to thank the Management Board of Ea So Nature Reserve, Eakar District, Dak Lak Province for providing mushroom samples for this study.

## Author contributions

**Conceptualization:** Trang Thi Thu Nguyen, Phu Tran Vinh Pham, Phu Chi Hieu Truong, Manh Hung Tran.

**Data curation:** Trang Thi Thu Nguyen, Phu Tran Vinh Pham, Viet An Thi Nguyen, Linh Thuoc Tran, Phu Chi Hieu Truong, Manh Hung Tran.

**Formal analysis:** Trang Thi Thu Nguyen, Trinh Thi Tuyet Nguyen, Linh Thuoc Tran, Phu Chi Hieu Truong, Manh Hung Tran.

**Funding acquisition:** Trang Thi Thu Nguyen, Manh Hung Tran.

**Investigation:** Phu Tran Vinh Pham, Viet An Thi Nguyen, Manh Hung Tran.

**Methodology:** Trang Thi Thu Nguyen, Trinh Thi Tuyet Nguyen, Linh Thuoc Tran, Phu Chi Hieu Truong, Manh Hung Tran.

**Project administration:** Manh Hung Tran.

**Resources:** Trang Thi Thu Nguyen, Trinh Thi Tuyet Nguyen, Linh Thuoc Tran, Phu Chi Hieu Truong, Manh Hung Tran.

**Software:** Trang Thi Thu Nguyen, Manh Hung Tran.

**Supervision:** Manh Hung Tran.

**Validation:** Manh Hung Tran.

**Visualization:** Phu Tran Vinh Pham, Viet An Thi Nguyen, Manh Hung Tran.

**Writing – original draft:** Trang Thi Thu Nguyen, Phu Tran Vinh Pham, Viet An Thi Nguyen, Phu Chi Hieu Truong, Manh Hung Tran.

**Writing – review & editing:** Trinh Thi Tuyet Nguyen, Linh Thuoc Tran, Manh Hung Tran.

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
