## [Decision Letter · Decision Letter 0]

9 Aug 2025

PONE-D-25-37431Classification and cytotoxic activity of Vietnamese *Ganoderma ellipsoideum* against human breast cancer MCF-7 cell linesPLOS ONE

Dear Dr. Tran,

Thank you for submitting your manuscript to PLOS ONE. After careful consideration, we feel that it has merit but does not fully meet PLOS ONE’s publication criteria as it currently stands. Therefore, we invite you to submit a revised version of the manuscript that addresses the points raised during the review process.

We look forward to receiving your revised manuscript.

Kind regards,

Muhammad Zeeshan Bhatti, Ph.D

Academic Editor

PLOS ONE

Journal Requirements:

https://www.researchsquare.com/article/rs-2421198/v1

https://www.tjnpr.org/index.php/home/article/view/5237

In your revision ensure you cite all your sources (including your own works), and quote or rephrase any duplicated text outside the methods section. Further consideration is dependent on these concerns being addressed.

“Scholarship Program of the Vingroup Innovation Foundation (VINIF), Vingroup Big Data Institute (VINBIGDATA) (grant number: code VINIF.2020.TS.68).”

“Trang Thi Thu Nguyen was funded by the Vingroup Joint Stock Company and supported by the PhD Scholarship Program of the Vingroup Innovation Foundation (VINIF), Vingroup Big Data Institute (VINBIGDATA) (grant number: code VINIF.2020.TS.68).”

“Scholarship Program of the Vingroup Innovation Foundation (VINIF), Vingroup Big Data Institute (VINBIGDATA) (grant number: code VINIF.2020.TS.68).”

Additional Editor Comments (if provided):

Thank you for submitting your manuscript, "Classification and cytotoxic activity of Vietnamese *Ganoderma ellipsoideum* against human breast cancer MCF-7 cell lines," to PLOS ONE. The paper has been peer-reviewed and requires a major revision before it can be considered for publication.

Below is a summary of the required revisions:

It has been noted that the manuscript's primary focus is on the identification and characterization of *Ganoderma ellipsoideum* using in silico methods, with very little experimental work on cytotoxicity. Please revise the title to better reflect the core hypothesis and content of your research.Please include a detailed description of the sample preparation and extraction methods in the section of Materials and MethodsTo validate you’re in silico results, please perform a Western blotting experiment for protein identificationCytotoxicity experiment results in the Tables 3 and 4 into a single table for clarity.The resolution of Figures 3, 4, 5, and 6 is too low. Please submit high-resolution versions of these images.The manuscript requires an extensive revision for grammar, syntax, and overall clarity. Please consider having it proofread by a native English speaker or a professional editing service.

We look forward to receiving your revised manuscript. Please ensure all comments are addressed in your resubmission.

Reviewers' comments:

Reviewer's Responses to Questions

**Comments to the Author**

1. Is the manuscript technically sound, and do the data support the conclusions?

Reviewer #1: Yes

Reviewer #2: Partly

2. Has the statistical analysis been performed appropriately and rigorously? 

Reviewer #1: Yes

Reviewer #2: Yes

3. Have the authors made all data underlying the findings in their manuscript fully available?

Reviewer #1: Yes

Reviewer #2: Yes

4. Is the manuscript presented in an intelligible fashion and written in standard English?

Reviewer #1: Yes

Reviewer #2: Yes

5. Review Comments to the Author

Reviewer #1: Overall, this is a solid and well-structured study. The authors present their methodology clearly and acknowledge key limitations—such as the use of a single breast cancer cell line, the need for experimental validation of docking results, and the lack of compound isolation—which demonstrates good scientific transparency. The conclusions are adequately supported, and the paper aligns with the scope and standards of PLOS ONE. Only a few minor editorial corrections are needed before publication. Minor edits for clarity and consistency:

• There's a small typo in the phrase “Maximum Likehood” — it should be corrected to “Maximum Likelihood.”

• In the Results section, it would be helpful to write out “Maximum Likelihood (ML)” the first time it's mentioned, even though it's already explained in the Methods.

• The phrase “anti cancer” should be written as one word: “anticancer.”

• Also, the word “basidicarps” appears to be misspelled — the correct term is “basidiocarps.”

Missing articles (a / the):

There are a couple of places where the article “a” is missing. For example:

• “...a previous study of Jiao and coworkers (2013)”

• “...as a potential inhibitor of anti-breast cancer targets”

Adding these will improve the flow and readability of the text.

These are all minor fixes that don’t impact the science in any way. Once these are addressed, I believe the manuscript will be ready for acceptance.

Reviewer #2: This study has several notable strengths. It reports *Ganoderma ellipsoideum* in Vietnam for the first time, confirmed through both morphological characterization and molecular phylogenetic analysis, thereby expanding the known distribution and taxonomy of the species. The work integrates taxonomy, in vitro cytotoxicity testing, and in silico molecular docking, linking biodiversity research to pharmacological potential. The focus on well-established breast cancer–related targets (HPA, MELK, CK2α, NUDT5) enhances biomedical relevance, and the identification of triterpenoids with predicted binding affinities exceeding that of tamoxifen highlights promising anticancer potential. The study also provides transparent sequence data deposition and applies statistical rigor, laying a strong foundation for future compound isolation, mechanistic work, and in vivo validation.

At the same time, several limitations should be acknowledged. Cytotoxicity was assessed only in a single breast cancer cell line (MCF-7) without comparison to normal cells, preventing evaluation of cancer selectivity. The findings are based solely on in vitro assays and in silico predictions, with no in vivo validation or direct biochemical confirmation of the predicted targets. The active compounds were not isolated or quantified from the tested fractions, and their individual contributions to the observed cytotoxicity remain unverified. Chemical composition of extracts was not characterized, and results may vary with different extraction batches or environmental conditions of fungal growth. Docking and ADMET analyses were predictive only and may not reflect biological behavior in living systems. Finally, the study is based on specimens collected from a single location, which may limit generalizability to other populations.

Given these limitations, the conclusions should be presented more modestly, or supported by additional experiments in future work. Specifically, including multiple cancer cell lines, incorporating non-transformed control cells to assess selectivity, and performing direct biochemical confirmation of predicted molecular targets, such as via western blotting, would strengthen the link between molecular docking predictions and observed biological activity.

6. PLOS authors have the option to publish the peer review history of their article (what does this mean? ). If published, this will include your full peer review and any attached files.

**Do you want your identity to be public for this peer review?** For information about this choice, including consent withdrawal, please see our Privacy Policy .

Reviewer #1: No

Reviewer #2: No

---

## [Author Response · Author response to Decision Letter 1]

16 Oct 2025

Response Journal Requirements:

Answer: We editted that the manuscript met PLOS ONE's style requirements.

https://www.researchsquare.com/article/rs-2421198/v1

https://www.tjnpr.org/index.php/home/article/view/5237

Answer: We have cited two publications, listed as references 34 and 60.

3. Thank you for stating the following financial disclosure: “Scholarship Program of the Vingroup Innovation Foundation (VINIF), Vingroup Big Data Institute (VINBIGDATA) (grant number: code VINIF.2020.TS.68).” Please state what role the funders took in the study. If the funders had no role, please state: "The funders had no role in study design, data collection and analysis, decision to publish, or preparation of the manuscript." If this statement is not correct you must amend it as needed. Please include this amended Role of Funder statement in your cover letter; we will change the online submission form on your behalf.

Answer: Thanks, we revised the problem, please see our funding statement.

Answer:

“Trang Thi Thu Nguyen was funded by the Vingroup Joint Stock Company and supported by the PhD Scholarship Program of the Vingroup Innovation Foundation (VINIF), Vingroup Big Data Institute (VINBIGDATA) (grant number: code VINIF.2020.TS.68).”

“Scholarship Program of the Vingroup Innovation Foundation (VINIF), Vingroup Big Data Institute (VINBIGDATA) (grant number: code VINIF.2020.TS.68).”

Answer: We revised funding-related text from the manuscript.

Response to Additional Editor Comments

1. It has been noted that the manuscript's primary focus is on the identification and characterization of *Ganoderma ellipsoideum* using in silico methods, with very little experimental work on cytotoxicity. Please revise the title to better reflect the core hypothesis and content of your research.

Answer: Thank you very much for this insightful comment. We fully understand and appreciate your suggestion to align the title more closely with the core hypothesis and content of the study, particularly regarding the cytotoxic potential against MCF-7 breast cancer cells. However, in this manuscript, our primary objective was to identify, classify, and report the presence of *Ganoderma ellipsoideum* in Vietnam for the first time. Although this species has been described in other regions, its discovery in Vietnam represents a noteworthy contribution to the national mycological and pharmacological biodiversity.

The in vitro cytotoxicity assay presented here serves as a preliminary step to explore the anticancer potential of this fungal species. We acknowledge that further in-depth studies are needed to fully elucidate its bioactivity. These investigations are currently in progress and will be the focus of future publications. We hope that the current title, or a slightly adjusted one, can still reflect the scientific contribution and scope of this work, which emphasizes both the taxonomic characterization and the initial screening for cytotoxic activity.

2. Please include a detailed description of the sample preparation and extraction methods in the section of Materials and Methods.

Answer: We added a detailed description of the sample preparation and extraction methods in the section of Materials and Methods.

3. To validate you’re in silico results, please perform a Western blotting experiment for protein identification

Answer: Thank you for the valuable suggestion. We fully acknowledge that validating in silico results requires complementary experimental evidence, including Western blotting and even in vivo studies. However, at this stage, we are currently facing limitations in terms of laboratory equipment and research funding, which prevent us from performing these essential experiments. We sincerely hope to overcome these constraints in the near future and conduct further studies to clarify and confirm the potential cytotoxic effects of the compound against human breast cancer MCF-7 cells.

4. Cytotoxicity experiment results in the Tables 3 and 4 into a single table for clarity.

Answer: We merged Cytotoxicity experiment results in the Tables 3 and 4 into a single table with name "Table 3".

5. The resolution of Figures 3, 4, 5, and 6 is too low. Please submit high-resolution versions of these images.

Answer: We sincerely apologize for the low resolution of Figures 3, 4, 5, and 6. These images were captured directly from the original docking software outputs, and unfortunately, higher-resolution versions are currently unavailable due to software limitations. Therefore, we kindly request to retain the current versions of these figures in the revised manuscript.

6. The manuscript requires an extensive revision for grammar, syntax, and overall clarity. Please consider having it proofread by a native English speaker or a professional editing service.

Answer: Thank you very much for your suggestion regarding grammar and clarity. We have thoroughly revised the manuscript to correct typographical, grammatical, and syntactical issues. The updated version has been carefully proofread to improve the overall readability and language quality. Please review the revised main manuscript file for confirmation of these improvements.

Response to Reviewers

REVIEWER 1 COMMENT AUTHOR RESPONSE PAGE NUMER

There's a small typo in the phrase “Maximum Likehood” — it should be corrected to “Maximum Likelihood.” We corrected from “Maximum Likehood”to “Maximum Likelihood.”

10

In the Results section, it would be helpful to write out “Maximum Likelihood (ML)” the first time it's mentioned, even though it's already explained in the Methods.

We wrote out “Maximum Likelihood (ML)” the first time it’s mentioned in the Results section. 10

The phrase “anti cancer” should be written as one word: “anticancer.” We wrote the phrase “anti cancer” to one word “anticancer”. 2

the word “basidicarps” appears to be misspelled — the correct term is “basidiocarps.” We corrected the term “basidicarps” to “basidiocarps”. 11

Missing articles (a / the):

There are a couple of places where the article “a” is missing. For example:

• “...a previous study of Jiao and coworkers (2013)”

• “...as a potential inhibitor of anti-breast cancer targets”

We added "a" into

• “... previous study of Jiao and coworkers (2013)”.

• “...as potential inhibitor of anti-breast cancer targets”

12

18

REVIEWER 2 COMMENT AUTHORS RESPONSE PAGE NUMER

This study has several notable strengths. It reports *Ganoderma ellipsoideum* in Vietnam for the first time, confirmed through both morphological characterization and molecular phylogenetic analysis, thereby expanding the known distribution and taxonomy of the species. The work integrates taxonomy, in vitro cytotoxicity testing, and in silico molecular docking, linking biodiversity research to pharmacological potential. The focus on well-established breast cancer–related targets (HPA, MELK, CK2α, NUDT5) enhances biomedical relevance, and the identification of triterpenoids with predicted binding affinities exceeding that of tamoxifen highlights promising anticancer potential. The study also provides transparent sequence data deposition and applies statistical rigor, laying a strong foundation for future compound isolation, mechanistic work, and in vivo validation.

At the same time, several limitations should be acknowledged. Cytotoxicity was assessed only in a single breast cancer cell line (MCF-7) without comparison to normal cells, preventing evaluation of cancer selectivity. The findings are based solely on in vitro assays and in silico predictions, with no in vivo validation or direct biochemical confirmation of the predicted targets. The active compounds were not isolated or quantified from the tested fractions, and their individual contributions to the observed cytotoxicity remain unverified. Chemical composition of extracts was not characterized, and results may vary with different extraction batches or environmental conditions of fungal growth. Docking and ADMET analyses were predictive only and may not reflect biological behavior in living systems. Finally, the study is based on specimens collected from a single location, which may limit generalizability to other populations.

Given these limitations, the conclusions should be presented more modestly, or supported by additional experiments in future work. Specifically, including multiple cancer cell lines, incorporating non-transformed control cells to assess selectivity, and performing direct biochemical confirmation of predicted molecular targets, such as via western blotting, would strengthen the link between molecular docking predictions and observed biological activity. Thank you for your comments. We have added a paragraph on the study’s limitations to the Conclusion section.

"Although G. ellipsoideum was first reported in China in 2018 and has since been cultivated in Thailand, our study is the first to document its discovery and distribution in Vietnam, specifically within the Ea So Nature Reserve of Dak Lak Province. Due to the limited number of natural fruiting bodies collected, compound isolation from Vietnamese samples was not yet possible. Instead, we prepared 70% ethanol extracts and fractionated extracts (n-hexane, ethyl acetate, and water) to evaluate their cytotoxicity against MCF-7 human breast cancer cells. To our knowledge, this is also the first study worldwide to assess the in vitro cytotoxic effects of this mushroom species on human breast cancer cells. Notably, we further investigated the inhibitory potential of its triterpenoids using in silico models targeting four key proteins associated with breast cancer (HPA, MELK, CK2α, and NUDT5). These findings provide a valuable foundation for future research, including evaluations across additional breast cancer cell lines, compound isolation, in vivo experiments, and gene expression studies (e.g., Western blot) to comprehensively assess the anti-breast cancer potential of G. ellipsoideum."

22

---

## [Decision Letter · Decision Letter 1]

20 Oct 2025

Taxonomic Characterization and Cytotoxic Potential of Vietnamese *Ganoderma ellipsoideum* Against Human Breast Cancer MCF-7 Cells

PONE-D-25-37431R1

Dear Dr. Tran,

We’re pleased to inform you that your manuscript has been judged scientifically suitable for publication and will be formally accepted for publication once it meets all outstanding technical requirements.

Kind regards,

Muhammad Zeeshan Bhatti, Ph.D

Academic Editor

PLOS ONE

Additional Editor Comments (optional):

Reviewers' comments:

Reviewer's Responses to Questions

**Comments to the Author**

1. If the authors have adequately addressed your comments raised in a previous round of review and you feel that this manuscript is now acceptable for publication, you may indicate that here to bypass the “Comments to the Author” section, enter your conflict of interest statement in the “Confidential to Editor” section, and submit your "Accept" recommendation.

Reviewer #1: All comments have been addressed

Reviewer #2: All comments have been addressed

2. Is the manuscript technically sound, and do the data support the conclusions?

Reviewer #1: Yes

Reviewer #2: Yes

3. Has the statistical analysis been performed appropriately and rigorously? 

Reviewer #1: Yes

Reviewer #2: N/A

4. Have the authors made all data underlying the findings in their manuscript fully available?

Reviewer #1: Yes

Reviewer #2: Yes

5. Is the manuscript presented in an intelligible fashion and written in standard English?

Reviewer #1: Yes

Reviewer #2: Yes

6. Review Comments to the Author

Reviewer #1: The authors have addressed all comments clearly and thoroughly. The revisions improved the manuscript, and I have no further concerns.

Reviewer #2: Documenting the first confirmed occurrence of *Ganoderma ellipsoideum* in Vietnam, the study supports this claim with both morphological description and molecular phylogenetic analysis, thereby expanding the species’ known taxonomy and geographic range. By combining taxonomic work with in vitro cytotoxicity assays and in silico molecular docking, the research links biodiversity investigation to pharmacological discovery. Focusing on well-established breast cancer targets (HPA, MELK, CK2α, and NUDT5) strengthens the biomedical relevance, and the identification of triterpenoids whose predicted binding affinities exceed that of tamoxifen points to promising anticancer candidates. Public deposition of sequence data and the application of appropriate statistical methods enhance reproducibility and provide a solid platform for future compound isolation, mechanistic studies, and in vivo validation.

7. PLOS authors have the option to publish the peer review history of their article (what does this mean? ). If published, this will include your full peer review and any attached files.

**Do you want your identity to be public for this peer review?** For information about this choice, including consent withdrawal, please see our Privacy Policy .

Reviewer #1: No

Reviewer #2: No

---

## [Editor Report · Acceptance letter]

PONE-D-25-37431R1

PLOS ONE

Dear Dr. Tran,

I'm pleased to inform you that your manuscript has been deemed suitable for publication in PLOS ONE. Congratulations! Your manuscript is now being handed over to our production team.

Kind regards,

on behalf of

Dr. Muhammad Zeeshan Bhatti

Academic Editor

PLOS ONE